# GAUSSIANFLUENT: GAUSSIAN SIMULATION FOR DYNAMIC SCENES WITH MIXED MATERIALS

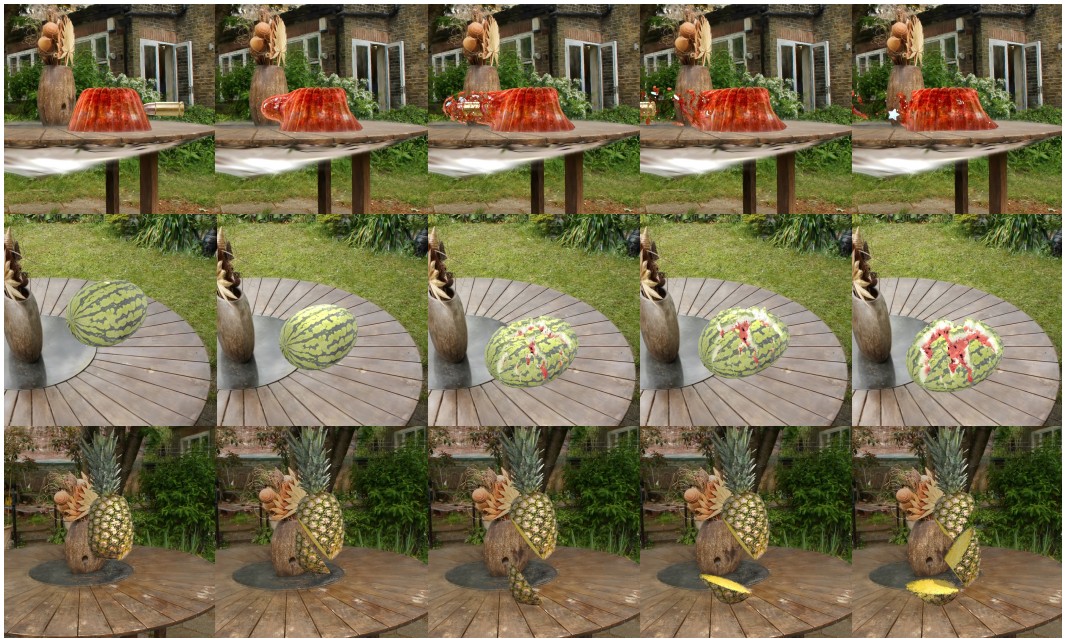

Figure 1: **Physical simulation of dynamic object states with 3D Gaussian Splatting. GaussianFluent** is capable of generating realistic internal texture, simulating and rendering complex object dynamics (*e.g.*, elastic deformation, fracture, and slicing) with mixed materials (*e.g.*, jelly with internal blue sugar penetrated by a rigid bullet in top row), in response to different lighting conditions.

## ABSTRACT

3D Gaussian Splatting (3DGS) has emerged as a prominent 3D representation for high-fidelity and real-time rendering. Prior work has coupled physics simulation with Gaussians, but it predominantly targets soft, highly deformable materials such as rubber and snow, leaving brittle fracture in objects like watermelons unresolved. This stems from two obstacles: the lack of a volumetric interior with coherent textures with GS representation, and the absence of fracture-aware simulation methods for Gaussians. To overcome these, we introduce **GaussianFluent**, a framework for realistic simulation and rendering of dynamic object states. First, it synthesizes consistent, photorealistic interiors by densifying internal Gaussians guided by generative models. Second, it integrates an optimized Continuum Damage Material Point Method (CD-MPM) to enable correct brittle fracture at real-time speeds. Finally, a Blinn-Phong model is used to shade the dynamically evolving fracture surfaces. Experiments show **GaussianFluent** delivers photo-realistic, real-time renderings of these state changes with structurally consistent interiors, highlighting its potential for downstream applications.

## 1 INTRODUCTION

3D Gaussian Splatting (3DGS) (Kerbl et al., 2023) has emerged as a prominent and highly effective technique for high-fidelity, real-time rendering of complex scenes and achieves state-of-the-art rendering quality. Despite its remarkable success, modeling dynamic scenes within the Gaussian Splat-

ting (GS) framework, especially the physics simulation of consistent evolution of multi-material objects, presents significant challenges. This difficulty stems from three primary issues.

First, as a surface-based method, GS inherently lacks representation of internal structures. Thus, the stress, inertia, and contact-force computations required for physically accurate solid-object simulation are undefined. It is also impossible for GS to realistically render the new surfaces exposed during the fracture process. Second, previous GS simulation methods, such as PhysGaussian (Xie et al., 2024), have largely targeted specific dynamics like elastic material. Simulation methods suitable for brittle fracture within the GS framework are still lacking. Existing point-cloud fracture methods (Wolper et al., 2019) are incompatible with GS; they lack a continuous return-mapping scheme, leading to physically implausible fracture dynamics, and their reliance on CPU-bound execution with limited parallelism imposes significant performance bottlenecks. Third, rendering dynamic lighting effects is challenging. Static GS reconstructions typically bake in lighting and shadows, which prevents their evolution during simulation. While advanced methods like Relightable 3DGS exist, they rely on pre-defined normals and physics properties, and thus cannot adapt to the evolving surfaces generated during fracture simulations.

To address these challenges, we introduce **GaussianFluent**, a framework to populate GS's interiors and simulate complex object dynamics such as brittle fracture and bullet impacts, with optionally dynamic lighting. Our key contributions are as follows:

- **Internal Texture Synthesis.** We propose a novel pipeline that synthesizes realistic and consistent internal structures and textures for GS by leveraging publicly available generative models, requiring no additional training data.

- **Optimized CD-MPM for GS.** We augment the current GS simulation framework with an optimized integration of CD-MPM, resolving instability issues in the previous CD-MPM algorithm and implementing GPU parallelism. This enables physics-plausible brittle fracture simulation with substantial real-time performance improvements.

- **Efficient Dynamic Lighting.** We implement a Blinn-Phong lighting model (Blinn, 1977), coupled with a normal estimation module. Its empirical formulation enables efficient lighting estimation for each evolving frame.

We validate **GaussianFluent** on a suite of challenging scenarios involving food, liquids, and fruits, where internal and external appearances differ significantly, and materials span brittle solids, viscoelastic gels, and soft tissues. Our experiments cover diverse topological changes, including dynamic fracturing, elastoplastic deformation, slicing, and high-velocity bullet impacts. Results show that our method effectively reconstructs structurally coherent internal GS primitives with realistic textures and achieves high-fidelity simulation and rendering of dynamic scenes, substantially outperforming existing methods.

## 2 RELATED WORK

### 2.1 DEFORMATION-PREDICTED DYNAMIC SCENES

Neural Radiance Fields (NeRF) (Mildenhall et al., 2021; Müller et al., 2022; Barron et al., 2021; 2023; 2022; Chen et al., 2022) and 3DGS (Kerbl et al., 2023; Yu et al., 2024; Huang et al., 2024a; Chen & Wang, 2024) have recently emerged as two prominent approaches for scene reconstruction, largely due to their ability to produce photo-realistic and efficient renderings. However, both methods primarily focus on static scenes and lack inherent support for modeling dynamic environments. To address this limitation, subsequent works incorporate deformation fields into neural radiance fields (Pumarola et al., 2021; Park et al., 2021; Tretschk et al., 2021) and Gaussian primitives (Wu et al., 2024; Yang et al., 2024b; Huang et al., 2024c; Wan et al., 2024; Liang et al., 2024; Luiten et al., 2024) to capture scene dynamics. Despite these advancements, existing approaches are typically limited to replaying observed motion trajectories rather than enabling further simulation or interaction, thereby restricting their generalization capability. Moreover, the modeling of motion in deformable Gaussians often lacks physically grounded constraints: each Gaussian is assigned an independent deformation vector without regard to physical plausibility, which can result in unrealistic or implausible dynamics.

## 2.2 Physics-Simulated Dynamic Scenes for Gaussian Splatting

3DGS is inherently compatible with MPM physics simulation frameworks, as its representation is composed of particle-like primitives, which provide a unified explicit foundation for both simulation and rendering. PhysGaussian (Xie et al., 2024) pioneers this direction by associating physical properties with Gaussian primitives and employing the Material Point Method (MPM) for physically based simulation. Subsequent works (Huang et al., 2024b; Zhang et al., 2024; Liu et al., 2024a) extend this framework by either learning physical properties from generative priors (Blattmann et al., 2023; Xing et al., 2024; Wang et al., 2023a; Lin et al., 2025), enabling automated physical parameter optimization. However, these methods still cannot model highly dynamic scenes, primarily due to the absence of simulation models suitable for brittle fracture. Furthermore, existing methods generally neglect the plausibility of internal textures that become visible when objects tear or break. FruitNinja (Wu & Chen, 2025) addresses internal texture generation for static GS reconstructions of fruits using a diffusion model fine-tuned on a self-collected dataset, which is costly and lacks generalizability. In addition, current simulation works (*e.g.* PhysGaussian) leave relighting unaddressed; existing lighting works like RelightableGS (Gao et al., 2024) and GS-Phong (He et al., 2024) typically rely on pre-computed, static surface normals, limiting their adaptability when fracture exposes new interior surfaces.

Building upon Continuum Damage Mechanics (CDM) (Simo & Ju, 1987; Matsuoka et al., 1999; Bourdin et al., 2000), we develop an optimized CD-MPM formulation for 3DGS that delivers brittle fracture on mixed-material objects, and pair it with (i) an internal texture filling pipeline and (ii) a dynamic lighting system for evolving Gaussians. The latter couples a Blinn-Phong model (Blinn, 1977) with training-free normal estimation, so newly fractured internal surfaces immediately obtain consistent shading and fragment-aware self-shadowing.

## 3 Method

We propose **GaussianFluent** to enable realistic simulation of dynamic scenes, particularly material fracture, within the 3DGS framework. The overall framework is shown in Figure 2. Our method first generates internal structures and textures for GS representations, followed by simulating fracture dynamics using an optimized CD-MPM framework. To achieve realistic rendering, we further incorporate dynamic lighting to accurately visualize newly exposed fracture surfaces.

### 3.1 Internal Filling for 3D Gaussian Splatting

#### 3.1.1 Internal Volume Initialization

Standard 3DGS primarily captures external surfaces, leaving interiors undefined, which is problematic for simulating interactions like cutting that expose internal structures. Our method first populates the interior volume and then textures it, as illustrated in Figure 2.

To initialize the internal volume, we first train an initial 3DGS model of the target object from multiview images. To prevent large Gaussians from straddling boundaries and ensure a clear exterior-interior separation (Liu et al., 2024c; Wu & Chen, 2025), we augment the standard rendering loss with a scale regularization:

$$L_{\text{total}} = 0.8\text{MSE} + 0.2\text{SSIM} + \lambda \sum_{i=1}^{N} ||\mathbf{s}_i||_2^2, \tag{1}$$

where $\mathbf{s}_i$ are the scale parameters of Gaussian $i$, and $\lambda$ controls regularization strength. This encourages smaller, more localized Gaussians, crucial for interior definition and relighting.

Next, we identify the object boundary as the high-density regions. Given a resolution $n$, we uniformly discretize the scene space into $n^3$ grids and compute the density field $d(x)$ for each grid center by accumulating contributions from its neighboring Gaussians $P$:

$$d(x) = \sum_{p \in P} \alpha_p \exp\Big( -\tfrac{1}{2}(x - x_p)^T \mathbf{A}_p^{-1}(x - x_p)\Big), \tag{2}$$

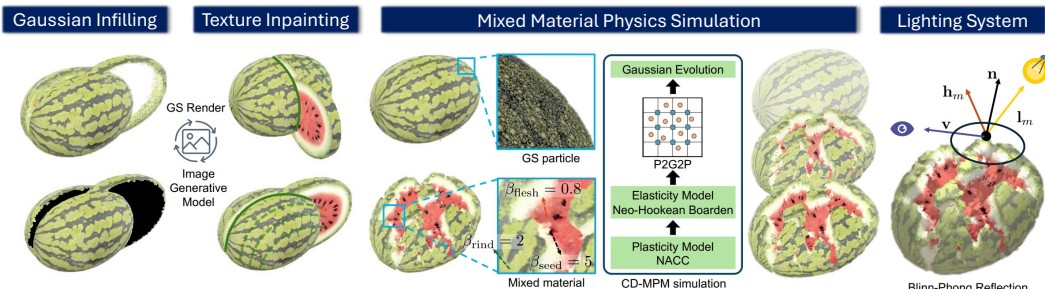

Figure 2: **Overview of GaussianFluent**. Our model first populates Gaussians in the internal volume and generates interior realistic texture with pretrained image generative models (Section 3.1). We then incorporate optimized CD-MPM simulation with mixed materials for Gaussian Splatting (Section 3.2) and introduce Blinn-Phong reflection in the rendering pipeline (Section 3.3).

where $\alpha_p$, $x_p$, and $\mathbf{A}_p$ denote the opacity, GS center, and covariance of Gaussian $p$, respectively. Grids with $d(x) \geq \tau_d$ are marked high-density; these high-density grids are extracted as the object boundary. New internal Gaussians are then initialized inside the enclosed volume, following prior practice by Xie et al. (2024).

This initial density-based filling can be imprecise, potentially creating Gaussians outside the true boundary due to sensitivities to surface geometry and threshold choice (Wu & Chen, 2025). To refine this, we perform an opacity-only optimization for all new internal Gaussians using the rendering loss by fixing other attributes. This drives the opacity of extraneous Gaussians to zero. Finally, we prune Gaussians with opacity $\alpha < \epsilon_\alpha$, resulting in a clean, well-defined solid volume representation suitable for subsequent texturing and simulation, as shown in Figure 3.

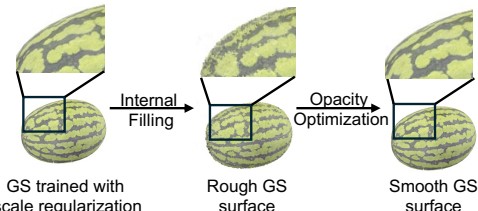

Figure 3: **Internal Gaussian filling and refinement.** The opacity optimization improves the smoothness of the GS surface after internal filling, beneficial for texture inpainting and simulation.

### 3.1.2 INTERNAL TEXTURE GENERATION

Once the interior volume is populated, assigning plausible internal textures is the next step. Generating multi-view and spatially coherent internal textures is a significant challenge due to scarce training data for object interiors (Poole et al., 2022; Liu et al., 2024b). Thus, we propose a training-free two-stage approach: an initial texture generation via single-view inpainting, followed by iterative multi-axis refinement, as illustrated in Figure 2.

**Coarse Texture Initialization** We first establish a coarse internal texture by uniformly slicing the object into 40 slices along the X-axis and inpainting each slice from its frontal viewpoint. For each slice, we render its initial appearance $\mathbf{C}_{\text{initial}}$ and an internal region mask $\mathbf{M}_{\text{init}}$. The masked region in $\mathbf{C}_{\text{initial}}$ is then inpainted using a generative model, *e.g.*, MVInpainter (Cao et al., 2024), guided by a text prompt $\mathcal{P}$, to produce the target image $\mathbf{C}_{\text{inpaint}}$. Each internal Gaussian $i$ whose 2D projection $\mathbf{u}_i$ falls within the inpainted region then samples its color $\mathbf{c}_i$ from $\mathbf{C}_{\text{inpaint}}$ using bilinear interpolation. Its zeroth-order spherical harmonic (SH) coefficient, $\mathbf{sh}_i^0$, is initialized as:

$$\mathbf{sh}_i^0 = \frac{\mathbf{c}_i - 0.5}{C_0}, \tag{3}$$

where the constant $C_0 = 1/(2\sqrt{\pi})$. Higher-order SH coefficients for these internal Gaussians are initialized to zero, ensuring an initially isotropic appearance derived from the inpainted texture.

**Iterative Texture Refinement** The single-view initialization, while a reasonable start, lacks consistency across the 3D internal structure and different viewing directions. To achieve comprehensive consistency, we iteratively refine the texture across all three primary axes (X, Y, and Z). This refinement is guided by text-prompted image inpainting using image generative models like Stable Diffusion XL (SD-XL) (Podell et al., 2023).

Inspired by the iterative corrective philosophy of SDS (Poole et al., 2022), we perform successive low-strength inpainting updates. The core refinement loop, repeated per iteration, consists of two main steps: 1) *Generative Inpainting of Slices*: We select 40 uniformly spaced slices along each of the X, Y, and Z axes (120 slices in total). For every slice, we render its axis-aligned orthographic view and an internal structure mask; these inputs are passed to SD-XL with an inpaint strength of 0.1, constraining denoising so that edits incrementally inject new internal details while remaining the global structure. 2) *Gaussian Optimization*: The newly inpainted 2D images from all 120 slices serve as optimization targets. The SH coefficients of the internal GS are optimized for 5 steps to minimize the rendering discrepancy against these inpainted images.

This two-step cycle is systematically repeated until the optimization loss converges or a maximum number of iterations is reached, ultimately yielding an internally consistent and detailed 3D texture. Our successive low-strength inpainting strategy produces sharp and realistic textures, in contrast to vanilla SDS, which leads to blurry and oversaturated results (Wang et al., 2023b; Alldieck et al., 2024; Lukoianov et al., 2024). Since the internal slices are co-dependent, with intersections on the orthogonal views, the iterative refinement drives the optimization toward tri-axial consistency.

### 3.2 CD-MPM in Gaussian Splatting with Mixed Materials

We extend the 3DGS simulation framework by incorporating the CD-MPM with support for mixed materials. Similar to PhysGaussian (Xie et al., 2024), each 3D Gaussian primitive in our framework is assigned physical properties, including mass, velocity, volume, and stress, and interacts with other particles via a background Eulerian grid. Our GPU parallelization implementation for efficient physical simulation is detailed in Appendix B.3.

**Initialization** We initialize covariances only for newly added interior Gaussians, assigning each a spherical covariance whose radius corresponds to its per-particle volume, *i.e.*, cell volume divided by the number of particles in the cell. The material parameters of the Gaussians, such as Young's modulus, Poisson's ratio, friction angle, mass density, etc., are manually defined following PhysGaussian and CD-MPM.

**GS Property Evolution with MPM** Let $\mathbf{X}$ denote the reference GS state before simulation, and $\mathbf{x}$ the state after simulation. Continuum mechanics describes motion via a time-dependent deformation map as follows:

$$\mathbf{x} = \boldsymbol{\varphi}(\mathbf{X}, t). \tag{4}$$

Here, $\boldsymbol{\varphi}$ represents the MPM simulation function. The deformation gradient $\mathbf{F}_p(t)$ is defined as

$$\mathbf{F}_p(t) = \frac{\partial \mathbf{x}}{\partial \mathbf{X}} = \frac{\partial \boldsymbol{\varphi}(\mathbf{X}, t)}{\partial \mathbf{X}}, \tag{5}$$

which encodes both local rigid deformation (rotation) and non-rigid deformation (stretch and shear). For each simulation step, we apply $\mathbf{F}_p(t)$ to the GS's covariance and spherical harmonics to achieve physics-plausible simulation results. For more details, please refer to Appendix A.

**Fracture Mechanism** We model brittle fracture by tracking the deformation $\mathbf{F}_p(t)$ of each GS. A softening law reduces its stress-generating capacity with increasing deformation. Fracture is not triggered by a sharp threshold but emerges when this capacity becomes negligible and fails to sustain internal forces. We decompose $\mathbf{F}_p(t)$ into rigid and non-rigid components; only the latter, comprising volumetric stretch $p$ and shear distortion $q$, contributes to fracture. A square under volumetric stretch becomes a scaled orthogonal rectangle, whereas pure shear turns it into an area-preserving parallelogram with skewed angles. The elastic, stress-generating region is defined by a yield surface $y(p, q) \leq 0$. With accumulating deformation, this surface contracts in the $(p, q)$-plane, dimin-

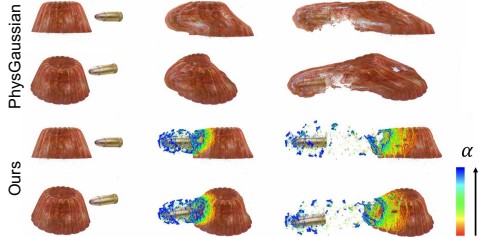

Figure 4: **A jelly-like material is shot with a bullet.** We compare our method with PhysGaussian to demonstrate the effectiveness of our simulation and visualize the damage variable $\alpha$.

ishing the sustainable elastic stress. Fracture occurs as this residual capacity vanishes. The Non-Associated Cam-Clay (NACC) model specifies this surface via the equation $y(p, q; p_0, \beta, M) = 0$. More specifically,

$$y(p, q; p_0, \beta, M) = q^2(1 + 2\beta) + M^2(p + \beta p_0)(p - p_0), \tag{6}$$

$$p_0 = K \sinh(\xi \max(-\alpha, 0)). \tag{7}$$

$\beta, M, K, \xi$ are all predefined hyperparameters, $p$ is the volumetric stretch magnitude, and $q$ is the shear magnitude. $\alpha$ is the key damage variable. At each step, we apply return mapping to enforce $y \leq 0$ and update $\alpha$ to evolve the yield surface $y$.

**Continuous Return Mapping**   At each step, a trial state $(p^{\mathrm{tr}}, q^{\mathrm{tr}})$ is formed and evaluated by the yield function $y^{\mathrm{tr}} = y(p^{\mathrm{tr}}, q^{\mathrm{tr}})$. Only the region where $y \leq 0$ is physically meaningful. Therefore, when $y > 0$, it is necessary to project $(p^{\mathrm{tr}}, q^{\mathrm{tr}})$ onto the ellipsoid such that $y = 0$. In CD-MPM, this projection process $R$ involves two possible cases:

1. Exterior pressures ($p^{\mathrm{tr}} \geq p_0$ or $p^{\mathrm{tr}} \leq -\beta p_0$): tip projection, where $p^{\mathrm{tr}} \geq p_0 \Rightarrow (p_0, 0)$, and $p^{\mathrm{tr}} \leq -\beta p_0 \Rightarrow (-\beta p_0, 0)$.

2. Interior pressures ($-\beta p_0 < p^{\mathrm{tr}} < p_0$): connect $(p^{\mathrm{tr}}, q^{\mathrm{tr}})$ to $(p_c, 0)$ with the ellipse center $p_c = \frac{-\beta p_0 + p_0}{2} = \frac{1 - \beta}{2} p_0$, where the intersection with the yield ellipse gives $(p_{\mathrm{new}}, q_{\mathrm{new}})$.

The connection to the fixed center $(p_c, 0)$ causes return-map discontinuities at $p = p_0$ and $p = -\beta p_0$. At the right boundary $p$, letting $q^{\mathrm{tr}} \to \infty$ shows a jump:

$$\lim_{\varepsilon \to 0^+} \lim_{q^{\mathrm{tr}} \to \infty} R(p_0 - \varepsilon, q^{\mathrm{tr}}) = (\frac{1 - \beta}{2} p_0, \frac{M(\beta + 1)}{2\sqrt{1 + 2\beta}} p_0), \tag{8}$$

$$\lim_{\varepsilon \to 0^+} \lim_{q^{\mathrm{tr}} \to \infty} R(p_0 + \varepsilon, q^{\mathrm{tr}}) = (p_0, 0). \tag{9}$$

We present a schematic diagram in Figure A1 to illustrate the discontinuity jump problem of this projection. Specifically, approaching $p \to p_0^-$ with $q^{\mathrm{tr}} \to \infty$ maps the trial state to the upper apex of the yield ellipse, whereas $p \to p_0^+$ maps it to the right tip $(p_0, 0)$. This jump triggers numerical instability: a machine-precision fluctuation $\delta$ about $p_0$ can map an identical geometric state to completely different return points. To resolve this instability, we regularize the projection by introducing a **dynamic point**, $(p'_c, 0)$, which smoothly adapts to the $(p, q)$ and ensures a continuous mapping. We define this new point as:

$$p'_c = p_c + \phi_k(p^{\mathrm{tr}})(p^{\mathrm{tr}} - p_c), \tag{10}$$

where $\phi_k(p^{\mathrm{tr}}) = \left| \frac{p^{\mathrm{tr}} - p_c}{p_0 - p_c} \right|^k$ and $p_0 - p_c$ is the semi-major axis of the ellipse.

This modified scheme can be regarded as an extension of the original approach, replacing the fixed point $(p_c, 0)$ with a dynamic point $(p'_c, 0)$. For any finite $k$, we have $\lim_{\varepsilon \to 0^+} p'_c(p_0 - \varepsilon) = p_0$, indicating continuity:

$$\lim_{\varepsilon \to 0^+} \lim_{q^{\mathrm{tr}} \to \infty} R(p_0 - \varepsilon, q^{\mathrm{tr}}) = \lim_{\varepsilon \to 0^+} \lim_{q^{\mathrm{tr}} \to \infty} R(p_0 + \varepsilon, q^{\mathrm{tr}}) = (p_0, 0). \tag{11}$$

Moreover, it recovers the original discontinuous scheme in the limit as $k \to \infty$, because for any $p_{\mathrm{tr}}$ in $(-\beta p_0, p_0)$, we have $\lim_{k \to \infty} p'_c = \lim_{k \to \infty}(p_c + \left| \frac{p^{\mathrm{tr}} - p_c}{p_0 - p_c} \right|^k) = p_c$. In our implementation, we choose $k = 2$, as it provides a robust and smooth projection. After projection, we compose $p$ and $q$ to obtain the $\mathbf{F}_p$. We define $J_{\mathrm{tr}} = \det \mathbf{F}_p^{\mathrm{tr}}$ and $J_{\mathrm{new}} = \det \mathbf{F}_p^{\mathrm{new}}$, and update the hardening parameter via $\alpha \leftarrow \alpha + \ln \left( \frac{J_{\mathrm{tr}}}{J_{\mathrm{new}}} \right)$. We present an example of $\alpha$ heatmap in Figure 4 to visualize the changes in $\alpha$ within the jelly. As the $\alpha$ value increases, the corresponding regions of the jelly undergo fracturing. Further details are provided in Appendix B.

**Mixed material simulation**   Unlike PhysGaussian, which assumes uniform material properties, our method supports more realistic and complex simulations by assigning different $\beta$ to various parts of an object, such as the seed, flesh, and rind of a watermelon. This requires segmenting

| Ours | $\beta = 0$ | $\beta = 0.6$ | $\beta = 1.2$ | $\beta = 5$ |

Figure 5: **Comparison between our mixed material modeling and fixed $\beta$ setting.** Our approach assigns distinct $\beta$ values, *i.e.*, 2, 0.6, and 5, to the rind, flesh, and seed, respectively. This yields more realistic simulation results compared to settings that apply a single, uniform $\beta$ value to the entire watermelon.

both external and internal structures through existing segmentation methods (Yang et al., 2024a; Liu et al., 2025), part-aware object generation (Yang et al., 2025; Zhang et al., 2025), or heuristics. For example, to realistically model a watermelon fracture, we assign $\beta$ values based on the color of the GS, *i.e.*, 5 to the black seeds, 0.6 to the red flesh, and 2 to the green rind. As shown in Figure 5, our mixed material approach produces more realistic results, whereas using a uniform material leads to visual artifacts and unnatural fracture patterns. The lollipop shattering scene in Figure 7 further demonstrates the cracks that PhysGaussian cannot generate.

### 3.3 RELIGHTING

Existing GS lighting methods, such as Relightable 3DGS (Gao et al., 2024) and GS-Phong (He et al., 2024), are not applicable to dynamic simulations. These methods are intended for static scenes and rely on multiple images captured under known lighting conditions to learn GS normals and other attributes. Rather than adopting the Physically Based Rendering (PBR) lighting model in Relightable 3DGS, which requires learning additional material attributes, *e.g.*, Fresnel parameters, for each GS, we employ the empirical Blinn-Phong reflection model, which only requires the normals for GS.

However, it is nontrivial to obtain GS normals using non-learning methods. As noted in Relightable 3DGS, numerical normal-estimation methods such as PCA are ill-suited to GS for two primary reasons: (i) GS particles are spatially sparse, and (ii) Gaussian centers, especially those with large kernels, are not tightly aligned with the visual surface. To overcome these issues, the regularization loss 1 we introduce in Section 3.1 promotes kernel densification and surface alignment, thereby enabling effective normal computation for each Gaussian splat using PCA.

**Blinn-Phong Reflection Model**    Once the normal $\mathbf{n}$ for each GS is computed, we apply the Blinn-Phong reflection model to determine its final color. For each Gaussian $i$ with center $\mathbf{p}_i$ and normal $\mathbf{n}_i$, we apply the Blinn-Phong reflection model using view direction $\mathbf{v}$ (from $\mathbf{p}_i$ to the camera) and, for each light $m$, light direction $\mathbf{l}_m$, distance $r_m$, and half vector $\mathbf{h}_m = (\mathbf{l}_m + \mathbf{v})/\|\mathbf{l}_m + \mathbf{v}\|_2$. The diffuse and specular terms are $D_m = \max(\mathbf{n}_i \cdot \mathbf{l}_m, 0)$ and $S_m = [\max(\mathbf{n}_i \cdot \mathbf{h}_m, 0)]^p$, with shininess exponent $p$. Let $\mathbf{c}_0$ be the base color, $\mathbf{I}_a$ the ambient light color, $\mathbf{I}_{L,m}$ the color of light $m$, and $T_{i,m}$ a per-light visibility term. Then

$$\mathbf{L}_i = \mathbf{c}_0 \odot \mathbf{I}_a + \sum_m T_{i,m} \left(\mathbf{c}_0 \odot \mathbf{I}_{L,m}\right) \frac{1}{r_m^2} \left(D_m + S_m\right), \tag{12}$$

where $\odot$ denotes element-wise multiplication.

This lighting framework allows us to effectively simulate complex scenes with multiple objects and dynamic light sources, as shown in Figure 1 and Figure A3. For example, in the latter figure, we present a scene of multiple fruits with dynamic lighting on a table. Such dynamic illumination and shadowing are crucial for achieving visually consistent and plausible renderings during simulation, where the evolution of shadows is not considered in PhysGaussian (Xie et al., 2024).

## 4 EXPERIMENT

We conduct experiments on both internal texture filling and physical simulation of dynamic scenes. For a more intuitive visualization of our results, we refer the reader to the ***supplementary videos***.

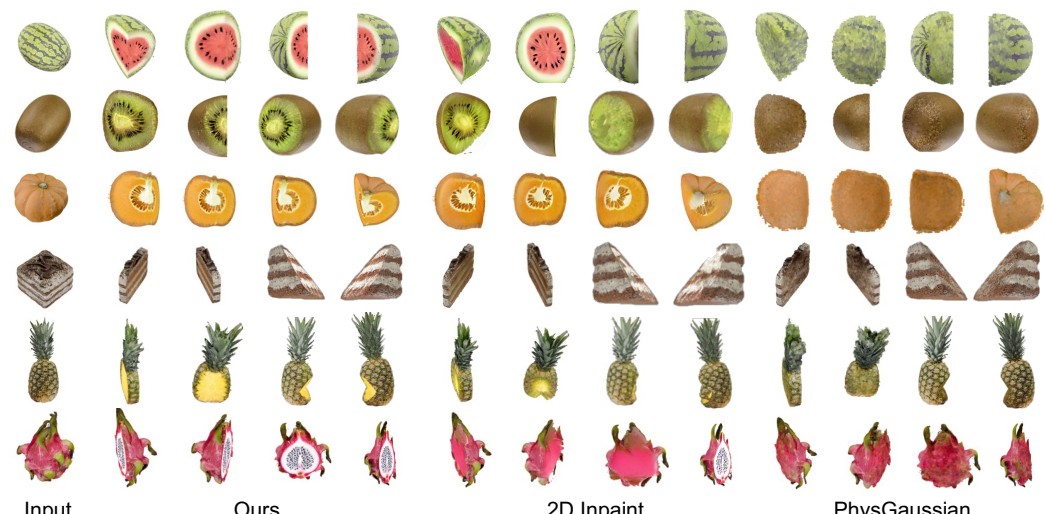

Input        Ours        2D Inpaint        PhysGaussian

Figure 6: **Qualitative comparison of internal texture filling.** Our method yields more realistic and consistent interior textures from GS rendering.

## 4.1 INTERNAL TEXTURE FILLING

We evaluate the quality of the generated interior texture, both quantitatively and qualitatively, against PhysGaussian and 2D Inpainting. We report CLIP scores (Radford et al., 2021) and conduct a user study, where participants are asked to select the best internal filling results. As presented in Table 1, our method achieves

Table 1: **Quantitative internal filling comparison.**

| Method | CLIP Score ↑ | User study ↑ |
|---|---|---|
| PhysGaussian | 22.3 | 3.57% (3/84) |
| 2D Inpainting | 30.1 | 25.00% (21/84) |
| Ours | 35.4 | 71.43% (60/84) |

the highest CLIP score, significantly outperforming PhysGaussian and 2D inpainting. These results indicate that the interior textures generated by our approach exhibit superior semantic consistency with the target descriptions. Prompt details are shown in Appendix C.

Figure 6 provides a qualitative comparison of rendered internal structures, and our method produces realistic and detailed results. For instance, the figure showcases the distinct seeds and flesh texture within a watermelon, a spherical cross-section of a kiwi that reveals its characteristic patterns, and an oblique slice through a cake displaying its clearly defined layers. These high-fidelity results stand in sharp contrast to those from PhysGaussian, which appear blurrier and less defined. Furthermore, while 2D inpainting can produce plausible individual slices, it fails to maintain 3D consistency across different views, resulting in unconvincing volumetric representations. In addition to static textures, our method achieves realistic dynamic rendering during simulation, capturing authentic material behavior under various conditions, as detailed in the next section.

## 4.2 PHYSICS SIMULATION FOR DYNAMIC SCENES

Quantitative evaluations of physics simulation further validate the performance of our method using both CLIP similarity scores and a perceptual user study, where participants are asked to choose the most realistic simulation outcome from our method and the baselines using the same form as Figure 7. As shown in Table 2,

Table 2: **Dynamic scene simulation comparison.**

| Method | CLIP Score ↑ | User study↑ |
|---|---|---|
| PhysGaussian | 12.2 | 3.84% (1/26) |
| OmniPhysGS | 13.1 | 7.69% (2/26) |
| Ours | 22.7 | 88.46% (23/26) |

our method achieves the highest CLIP similarity score and user preference, substantially outperforming PhysGaussian and OmniPhysGS. The perceptual study demonstrates that our results align with the users' understanding of realistic dynamic evolution and conform to intuitive physical commonsense. The higher CLIP score affirms that our simulation outcomes are not only visually more convincing but also semantically more accurate.

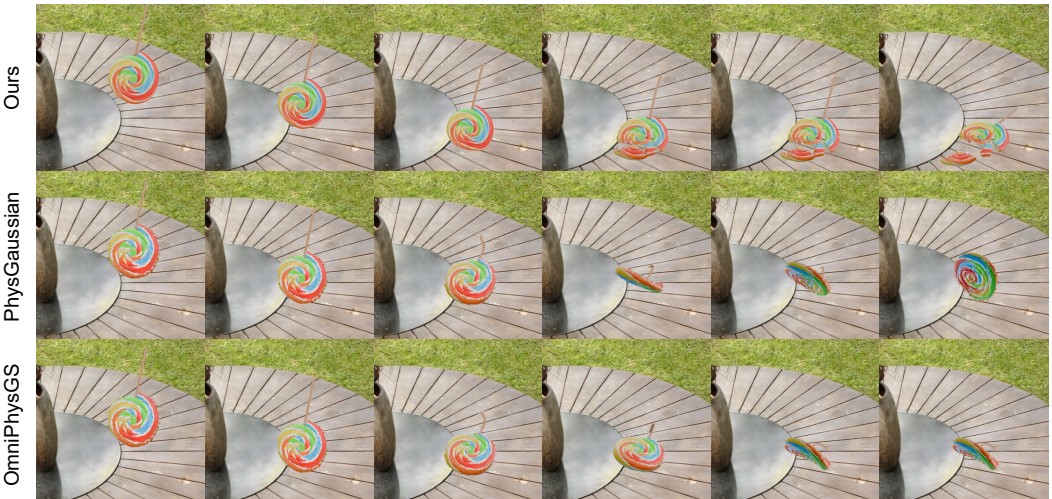

Figure 7: **Qualitative comparison of object state simulation.** We present a comparison for a lollipop, where our result correctly simulates its fracture of mixed materials, outperforming PhysGaussian and OmniPhysGS.

**Diverse Object Simulation**   We conduct an extensive series of qualitative experiments, as shown in Figures 1 and 7, and Figures A2 and A3 in *Appendix*, to further substantiate the broad applicability and robustness of our framework for various objects with significantly different material properties. The set included highly elastic materials like jelly, sliceable fruits such as pineapples and kiwis, brittle objects like watermelons, fluids including milk, and granular structures like sandcastles. They provide compelling visual evidence of our model's capabilities under different lighting conditions in diverse scenarios. For example, the top example in Figure 1 illustrates the deformation of a jelly when struck by a bullet, highlighting not only its elastic response but also the detailed internal rigid sugar expulsion. More examples are shown in the supplementary video.

**Mixed-Material Physics Simulation**   Our method is capable of simulating complex fracture and deformation, particularly for objects with different material responses. Figure 7 highlights this with a challenging scenario: a lollipop shatters on impact while its wooden stick remains intact. This ability to model mixed-material physics simulations that are visibly more detailed and realistic than those from prior work. This is also demonstrated in our simulation of a falling watermelon (Figure 1, Figure 5), where the internal seed and flesh remain distinctly separate.

## 5   CONCLUSION AND DISCUSSION

In this paper, we introduce **GaussianFluent**, a novel framework for physically plausible and realistic simulations of dynamic scenes with 3D Gaussian Splatting, including material fracture and behaviors of mixed materials. Our core contributions include a method for internal structure texture synthesis, an adapted CD-MPM for efficient physics simulation, and a dynamic lighting system for rendering evolving fracture surfaces. This integration allows **GaussianFluent** to simulate complex events like shattering, deformation, fluid splashing, cutting, and granular collapse with high visual fidelity directly within the GS representation, as demonstrated by our diverse qualitative results. The ability to model, simulate, and render dynamic scenes paves the way for more applications involving dynamic and interactive virtual worlds.

**Limitations and Future Direction**   To further enhance the applicability and generalization of physical simulation in the GS framework, we point out several directions for future work. Firstly, enhancing physical accuracy and versatility could be achieved by incorporating a broader range of constitutive models and exploring simulation techniques better suited for specific phenomena like fluids. Secondly, the current physical parameters are manually set; automating this process through inverse rendering or learning-based approaches would reduce tuning efforts and could improve simulation fidelity. Future research could also focus on scalability for extremely complex scenes, more intricate multi-physics interactions, and integrating learning for predictive simulation.

## 6 REPRODUCIBILITY STATEMENT

We are committed to ensuring the full reproducibility of our work on **GaussianFluent**. To this end, we will release the complete source code, which includes our implementations of internal structure texture synthesis, the adapted CD-MPM for physics simulation, and the dynamic lighting system for rendering fracture surfaces. The curated dataset, derived from Objaverse, along with the Blender scripts used to render the training images, will also be made publicly available. We believe these comprehensive resources will enable the community to verify our findings and build upon the **GaussianFluent** framework for future research. Please refer to the Appendix C for specific implementation details.

## 7 ETHICS STATEMENT

We have thoroughly reviewed the ICLR Code of Ethics and confirm that all aspects of our work comply with established academic ethical standards. Our research does not involve human or animal subjects, nor does it contain any potentially harmful insights, methodologies, or applications. We do not anticipate any issues related to discrimination, bias, fairness, privacy, or security. Furthermore, our work adheres to all relevant legal and research integrity requirements, and we are confident that it aligns with the principles outlined in the ICLR Code of Ethics.

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

APPENDIX

# A MATERIAL POINT METHOD

**Overview**  We use an explicit MPM. Particles (also the 3D Gaussian splats) carry

$$m_p,\ V_p^0,\ \mathbf{X}_p,\ \mathbf{x}_p,\ \mathbf{v}_p,\ \mathbf{F}_p,\ \mathbf{A}_p,\ \mathbf{a}_p,\ \sigma_p,\ \mathbf{C}_p. \tag{A1}$$

In summary, given the 3D GS of a static scene $\{X_p, A_p, \sigma_p, C_p\}$, we use simulation to dynamize the scene by evolving these Gaussians to produce dynamic Gaussians $\{x_p(t), a_p(t), \sigma_p, C_p\}$. Here, $\mathbf{X}_p$ is the initial position, while $\mathbf{x}_p$ is the current position that evolves over time with velocity $\mathbf{v}_p$. Furthermore, $\mathbf{A}_p$ is the static covariance of the initial Gaussian; the dynamic covariance $\mathbf{a}_p$ is derived at each step; $\mathbf{F}_p$ is the deformation gradient used to calculate $\mathbf{a}_p$; and the opacity $\sigma_p$ and SH coefficient magnitudes $\mathbf{C}_p$ are considered time-invariant.

## A.1 THE MATERIAL POINT METHOD (MPM) ALGORITHM STEPS

The Material Point Method (MPM) algorithm iteratively transfers data between particles and a background grid. A single time step can be broken down into the following three main stages.

### A.1.1 PARTICLE-TO-GRID TRANSFER (P2G)

In the first stage, information is transferred from the Lagrangian particles to the nodes of the Eulerian grid. This process, known as rasterization, effectively creates a grid-based snapshot of the continuum's state. For each particle $p$, its mass $m_p$ and momentum $\mathbf{p}_p = m_p \mathbf{v}_p$ are interpolated and added to the surrounding grid nodes $i$. This is done using interpolation functions $N_{ip}$ (also known as shape functions), which depend on the particle's position relative to the grid.

The nodal mass $m_i$ and nodal momentum $\mathbf{p}_i$ are computed as follows:

$$m_i = \sum_p m_p N_{ip} \tag{A2}$$

$$\mathbf{p}_i = \sum_p m_p \mathbf{v}_p N_{ip}. \tag{A3}$$

From the nodal momentum and mass, the initial nodal velocity is found: $\mathbf{v}_i = \mathbf{p}_i / m_i$, provided $m_i > 0$.

### A.1.2 GRID UPDATE

This stage contains the core physics computations, which are performed entirely on the grid. First, forces acting on each grid node are calculated. These forces are typically composed of two parts:

- **Internal forces** $\mathbf{f}_i^{\text{internal}}$, which arise from the material's stress. These are computed by transferring particle stress information (derived from the deformation gradient $\mathbf{F}_p$) back to the grid.

- **External forces** $\mathbf{f}_i^{\text{external}}$, such as gravity or user-defined interactions.

The total force on a node is $\mathbf{f}_i = \mathbf{f}_i^{\text{internal}} + \mathbf{f}_i^{\text{external}}$.

With the total force, the grid node velocities are updated over the time step $\Delta t$ using an explicit time integration scheme (e.g., Forward Euler):

$$\mathbf{v}_i^{n+1} = \mathbf{v}_i^n + \Delta t \frac{\mathbf{f}_i}{m_i}. \tag{A4}$$

Boundary conditions, such as collisions with obstacles, are also enforced on the grid during this stage by modifying the nodal velocities.

### A.1.3 GRID-TO-PARTICLE TRANSFER (G2P)

Finally, the updated kinematic information is transferred from the grid back to the particles. This stage, often called the "gather" step, updates the Lagrangian particles' state using the newly computed fields on the Eulerian grid, preparing them for the next time step. This process involves updating each particle's velocity, its deformation gradient, and finally its position.

First, the particle's velocity $\mathbf{v}_p$ is updated by interpolating the new velocities $\mathbf{v}_i^{n+1}$ from the surrounding grid nodes. This is essentially a weighted average, using the same interpolation functions $N_{ip}$ as the P2G step:

$$\mathbf{v}_p^{n+1} = \sum_i \mathbf{v}_i^{n+1} N_{ip}. \tag{A5}$$

This update can be a pure Particle-In-Cell (PIC) update, or it can be combined with the particle's previous velocity in a FLIP (Fluid-Implicit-Particle) scheme to reduce numerical dissipation.

Simultaneously, the particle's deformation gradient $\mathbf{F}_p$, which tracks the local rotation and strain of the material, must also be updated. This is done by first computing the velocity gradient $\nabla \mathbf{v}$ at the particle's position, which is also interpolated from the grid node velocities:

$$\nabla \mathbf{v}_p = \sum_i \mathbf{v}_i^{n+1} \nabla N_{ip}^T. \tag{A6}$$

This gradient is then used to advance the deformation gradient forward in time:

$$\mathbf{F}_p^{n+1} = (\mathbf{I} + \Delta t \, \nabla \mathbf{v}_p) \, \mathbf{F}_p^n, \tag{A7}$$

where $\mathbf{I}$ is the identity matrix. This update is crucial for correctly computing material stress in the next time step.

Lastly, with the new velocity $\mathbf{v}_p^{n+1}$ computed, the particle's position $\mathbf{x}_p$ is updated as:

$$\mathbf{x}_p^{n+1} = \mathbf{x}_p^n + \Delta t \, \mathbf{v}_p^{n+1}. \tag{A8}$$

Once all particles have been updated, the information on the background grid is no longer needed and is typically reset or discarded. The simulation is now ready to begin the next time step with the P2G phase.

### A.2 EVOLUTION OF 3D GAUSSIAN PROPERTIES VIA CONTINUUM MECHANICS

This approach outlines a method for animating 3D GS by treating them as discrete particles within a physics-based system governed by continuum mechanics. The primary goal is to evolve a static scene, defined by initial properties, into a dynamic state for rendering.

The evolution of the key Gaussian properties for each time step is as follows:

- **Position Evolution (Mean):** The Gaussian's center, or mean, is its world-space position $\mathbf{x}_p$. This is updated using the particle's velocity $\mathbf{v}_p$, which is determined by the physical simulation, via explicit time integration:

$$\mathbf{x}_p^{n+1} = \mathbf{x}_p^n + \Delta t \, \mathbf{v}_p. \tag{A9}$$

- **Shape Evolution (Covariance):** The dynamic world-space covariance $\mathbf{a}_p$, which defines the Gaussian's shape and size, is computed directly from the deformation gradient $\mathbf{F}_p$. The deformation gradient describes the local deformation of the material around the particle. It maps the initial, undeformed shape (defined by the material-space covariance $\mathbf{A}_p$) to its current, deformed configuration:

$$\mathbf{a}_p(t) = \mathbf{F}_p(t)\mathbf{A}_p\mathbf{F}_p(t)^T. \tag{A10}$$

- **Orientation Evolution (for Rendering):** To correctly render anisotropic appearances (e.g., using Spherical Harmonics), the particle's orientation must be tracked. The rotation component $\mathbf{R}_p$ is extracted from the deformation gradient, typically via polar decomposition ($\mathbf{F}_p = \mathbf{R}_p\mathbf{S}_p$). This rotation is then applied to the appearance model during rendering.

- **Time-Invariant Properties:** Visual attributes such as opacity $\sigma_p$ and material-space appearance coefficients (e.g., Spherical Harmonics, $\mathbf{C}_p$) are considered intrinsic material properties. They are typically held constant throughout the simulation.

# B  FRACTURE MECHANISM WITH CONTINUUM DAMAGE MATERIAL POINT METHOD

## B.1  INTRODUCTION OF CD-MPM

The yield surface serves as a dividing boundary in stress space: inside it, the material response is elastic; at the boundary plastic yielding begins; any trial state predicted beyond this boundary is reconciled by returning it to a suitable point on the boundary in accordance with ideal plasticity. As mentioned above, the yield surface of CD-MPM is defined as:

$$y(p,q) = (1+2\beta)\, q^2 + M^2(p+\beta p_0)(p-p_0) = 0. \tag{A11}$$

If $(p,q)$ lies in the elastic domain where $y \leq 0$, no plastic correction is applied.

$$(p_c, q_c) = \left(\frac{1-\beta}{2}p_0,\, 0\right) \tag{A12}$$

$$y_{tr} = y(p_{tr}, q_{tr}) \tag{A13}$$

$$J_E(p) = \sqrt{-\frac{2p}{\kappa} + 1} \tag{A14}$$

Here $p_c, q_c$ identify the center of the yield ellipsoid ($y = 0$); $p_{tr}, q_{tr}$ is the uncorrected trial stress state produced at simulation step $n$; $J_E$ is the determinant of the elastic deformation gradient (elastic volume ratio); $\kappa$ is the Bulk Modules; and $p_{n+1}, q_{n+1}$ is the state after applying the return mapping $R$:

$$R(p_{n+1}, q_{n+1}) = \begin{cases} (p_{tr}, q_{tr}), & y_{tr} \leq 0 & \text{(Elastic)} \\ (p_0, 0), & y_{tr} > 0 \,\wedge\, p_{tr} > p_0 & \text{(Case 1: upper tip projection)} \\ (-\beta p_0, 0), & y_{tr} > 0 \,\wedge\, p_{tr} < -\beta p_0 & \text{(Case 2: lower tip projection)} \\ (p_x, q_x), & y_{tr} > 0 \,\wedge\, -\beta p_0 \leq p_{tr} \leq p_0 & \text{(Case 3: center–trial line intersection)} \end{cases} \tag{A15}$$

Here $y_{tr} = y(p_{tr}, q_{tr})$. If $y_{tr} \leq 0$, the trial point lies in the elastic domain and is accepted unchanged: $(p_{n+1}, q_{n+1}) = (p_{tr}, q_{tr})$. If $y_{tr} > 0$ and $p_{tr} > p_0$, the trial point lies beyond the positive $p$-axis tip and is projected to the upper tip $(p_0, 0)$. If $y_{tr} > 0$ and $p_{tr} < -\beta p_0$, it lies beyond the negative tip and is projected to $(-\beta p_0, 0)$. Otherwise ($y_{tr} > 0$ with $-\beta p_0 \leq p_{tr} \leq p_0$), we join the center $(p_c, q_c)$ and the trial point $(p_{tr}, q_{tr})$; the intersection of this line segment with the yield ellipsoid $y(p, q) = 0$ defines $(p_x, q_x)$, and we set $(p_{n+1}, q_{n+1}) = (p_x, q_x)$. Besides $p, q$, we also update $\alpha$ and $J_E$ as below:

$$\alpha_{n+1} = \alpha_n + \begin{cases} 0, & y_{tr} \leq 0 \\ \log\big(J_{E,tr}/J_{E,n+1}\big), & y_{tr} > 0 \end{cases}, \tag{A16}$$

with

$$J_{E,n+1} = \begin{cases} J_E(p_0), & \text{Case 1} \\ J_E(-\beta p_0), & \text{Case 2} \\ J_E(p_x), & \text{Case 3} \end{cases} \tag{A17}$$

## B.2  ADAPTED CONTINUOUS RETURN MAPPING

However, this piecewise return mapping is discontinuous at the right tip $p = p_0$. Consider trial states with $y_{tr} > 0$ and very large shear measure $q_{tr} \to \infty$. Take two sequences with $p_{tr} = p_0 - \varepsilon$ and $p_{tr} = p_0 + \varepsilon$ ($\varepsilon > 0$). For $p_{tr} = p_0 - \varepsilon$, the algorithm falls into the "center–trial line intersection" branch; as $q_{tr} \to \infty$ the direction from the center $(p_c, 0)$, with $p_c = \frac{1-\beta}{2}p_0$, to the trial point becomes vertical, so the mapped point tends to the upper apex of the yield ellipsoid $y(p, q) = 0$, namely,

$$(p_c, q_{\text{apex}}) = \left(\tfrac{1-\beta}{2}p_0,\, \tfrac{M(\beta+1)}{2\sqrt{1+2\beta}}\, p_0\right). \tag{A18}$$

Subsequently letting $\varepsilon \to 0^+$ leaves this limit unchanged. In contrast, for $p_{tr} = p_0 + \varepsilon$, the "upper tip projection" branch is invoked and the image is the tip $(p_0, 0)$, retaining the volumetric (tensile) part and removing shear. Thus,

$$\lim_{\varepsilon \to 0^+} \lim_{q_{tr} \to \infty} R(p_0 - \varepsilon, q_{tr}) = \left( \tfrac{1-\beta}{2} p_0, \, \tfrac{M(\beta+1)}{2\sqrt{1+2\beta}} p_0 \right), \lim_{\varepsilon \to 0^+} \lim_{q_{tr} \to \infty} R(p_0 + \varepsilon, q_{tr}) = (p_0, 0), \quad \text{(A19)}$$

showing a directional jump: one limit preserves (essentially) shear while the other preserves only the volumetric extension. And even some small $q$ such as $q = p_0$ will also occur jumps like this.

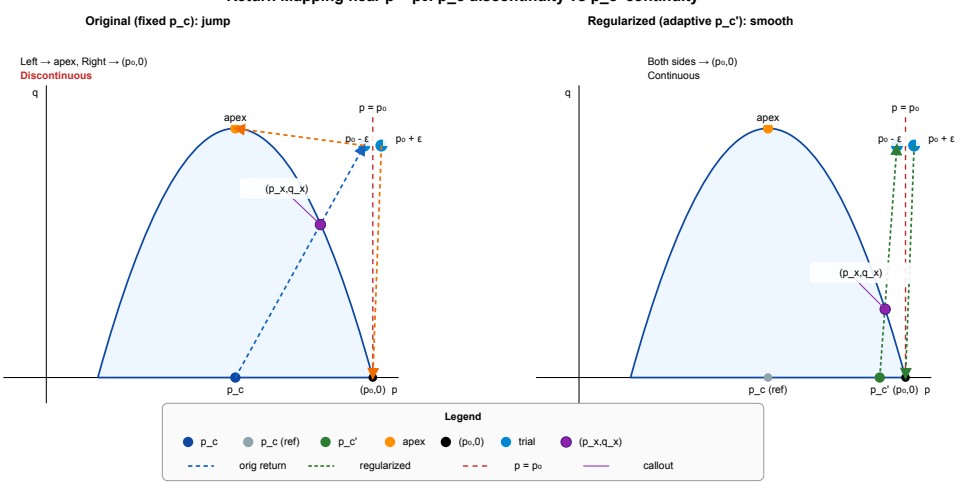

Figure A1: Comparison of two return mapping kinds.

To remove both the numerical instability and the physical ambiguity at the tip, we replace the interior $(p_{tr} \in [-\beta p_0, p_0])$ center–line branch with a normal closest-point return: solve $(p_{n+1}, q_{n+1}) = (p_{tr}, q_{tr}) - \Delta\lambda \nabla y(p_{n+1}, q_{n+1})$, $y(p_{n+1}, q_{n+1}) = 0$, $\Delta\lambda \geq 0$. Outside this interval, we still project to the nearest tip. This yields a continuous mapping and a well-defined consistent tangent.

We modify only the interior plastic branch with $-\beta p_0 \leq p_{tr} \leq p_0$. Introduce a $k$-dependent pseudo-center on the $p$-axis:

$$L_p = p_0 - p_c > 0, \qquad \phi_k = \left| \frac{p_{tr} - p_c}{L_p} \right|^k \in [0, 1], \qquad (p'_c, q'_c) = \big( p_c + \phi_k(p_{tr} - p_c), \, 0 \big). \quad \text{(A20)}$$

In Case 3 we replace the fixed center $(p_c, 0)$ by $(p'_c, 0)$, draw the line through $(p'_c, 0)$ and the trial point $(p_{tr}, q_{tr})$, and take its intersection with the yield surface $y = 0$ as the updated stress, as shown in Figure A1. All other cases are unchanged. For any finite $k$ the return mapping is continuous, because as $p_{tr} \to p_0^-$ we have $\phi_k \to 1$ and thus $p'_c \to p_{tr}$, so the update approaches the right tip smoothly. For any fixed interior $p_{tr} < p_0$, $\phi_k \to 0$ as $k \to \infty$, giving $p'_c \to p_c$ and recovering the original (unmodified) branch. Hence $k$ provides a homotopy from a continuous regularized mapping (finite $k$) back to the original formulation ($k \to \infty$).

### B.3 GPU Parallelization

We achieve a substantial performance improvement by porting the CPU-bound CD-MPM algorithm to the GPU. **Our implementation reduces simulation times from 4 minutes per frame to a single second**. This is accomplished through a complete framework reimplementation that leverages the NVIDIA Warp library to parallelize the core simulation loop. Unlike the original CPU-only method, our GPU-native approach enables the simulation of far more complex scenes in interactive time.

## C Experiment details

All experiments are conducted on a GPU capable of 52.22 TFLOPS (FP32) and approximately 103 Tensor TFLOPS (FP16). These simulations typically consume around 10 GB of VRAM, with peak

Table A1: **Parameters and Timings.** Seconds per frame (s/frame) is an average. All performance metrics were obtained from experiments conducted on a GPU delivering 103 Tensor TFLOPS at FP16 precision.

| Example | s/frame | $\Delta t_{frame}$ | $\Delta x$ | $\Delta t_{step}$ | $N$ | $\rho$ | $E$ | $\nu$ | NACC-$(\alpha_0, \beta, \xi, M)$ |
|---|---|---|---|---|---|---|---|---|---|
| watermelon | 3.56 | 1/50 | $3 \times 10^{-3}$ | $1 \times 10^{-4}$ | 27M | 2 | 2000/1000/$1 \times 10^4$ | 0.38 | (-0.04, 2/0.6/5, 2, 2.36) |
| jelly | 0.39 | 1/500 | $3 \times 10^{-3}$ | $1 \times 10^{-5}$ | 1M | 2 | 2000 | 0.45 | (-0.5, 1, 2, 2.36) |
| pumpkin | 5.12 | 1/50 | $3 \times 10^{-3}$ | $1 \times 10^{-4}$ | 27M | 2 | 4000 | 0.40 | (-0.04, 1, 2, 2.36) |
| kiwi | 1.58 | 1/50 | $1 \times 10^{-2}$ | $1 \times 10^{-4}$ | 1M | 2 | 2000 | 0.42 | (-0.04, 1, 2, 2.36) |
| pineapple | 1.16 | 1/50 | $1 \times 10^{-2}$ | $1 \times 10^{-4}$ | 1M | 2 | 5000 | 0.39 | (-0.04, 1, 2, 2.36) |
| dragonfruit | 2.27 | 1/50 | $1 \times 10^{-2}$ | $1 \times 10^{-4}$ | 1M | 2 | 2000 | 0.42 | (-0.04, 1, 2, 2.36) |
| tosta | 3.18 | 1/50 | $5 \times 10^{-3}$ | $1 \times 10^{-4}$ | 8M | 2 | 2000 | 0.38 | (-0.1, 1, 2, 2.36) |
| sandcastle | 2.09 | 1/50 | $1 \times 10^{-2}$ | $1 \times 10^{-4}$ | 8M | 2 | 50 | 0.05 | (-0.04, 0.01, 1, 2.36) |

usage not exceeding 16 GB. Detailed timings and material parameters are provided in Table A1. For the NACC model, the parameter $\beta$ is adjusted to differentiate the material properties of various components, while the initial parameter $\alpha_0$ is maintained uniformly for all particles within an object.

For coarse texture generation, MVInpainter (Shi et al., 2023) is selected over IP-Adapter (Ye et al., 2023) and MVDream (Shi et al., 2023) due to its ability to maintain color consistency across different viewing axes. Subsequently, SD-XL was employed for fine texture generation, owing to its enhanced performance in generating detailed interior textures compared to IP-Adapter.

For the user study, we prepare eight distinct objects: watermelon, cake, jelly, pumpkin, bread, kiwi, dragonfruit, and pineapple. We then conduct two separate evaluations. To assess the quality of the interior filling, we recruit 21 participants, collecting a total of $8 \times 21 = 168$ ratings. Separately, to evaluate the simulation dynamics, 26 participants are recruited, providing a total of $8 \times 26 = 208$ ratings.

We use two types of prompts:

- **For interior filling**: We explicitly instruct GPT to generate an inpainting prompt in the form "a slice of [object]." For example, GPT produces the following for a watermelon: "A realistic and detailed drawing of the juicy red flesh and black seeds of a watermelon slice."

- **For CLIP-score evaluation**: To evaluate the plausibility of the final scene, we have human annotators write prompts that describe the overall event, for example, "A watermelon dropped and shattered on a table," and "Slices of a [object] landing on a table."

## D    USE OF LARGE LANGUAGE MODELS (LLMS)

We used a large language model solely as a writing aid to improve the clarity, grammar, and overall readability of the manuscript. Its role was limited to polishing the language and refining sentence structure, without contributing to research ideation, experimental design, or data analysis. All technical ideas, methods, results, and conclusions are entirely the work of the authors, and we take full responsibility for the final content.

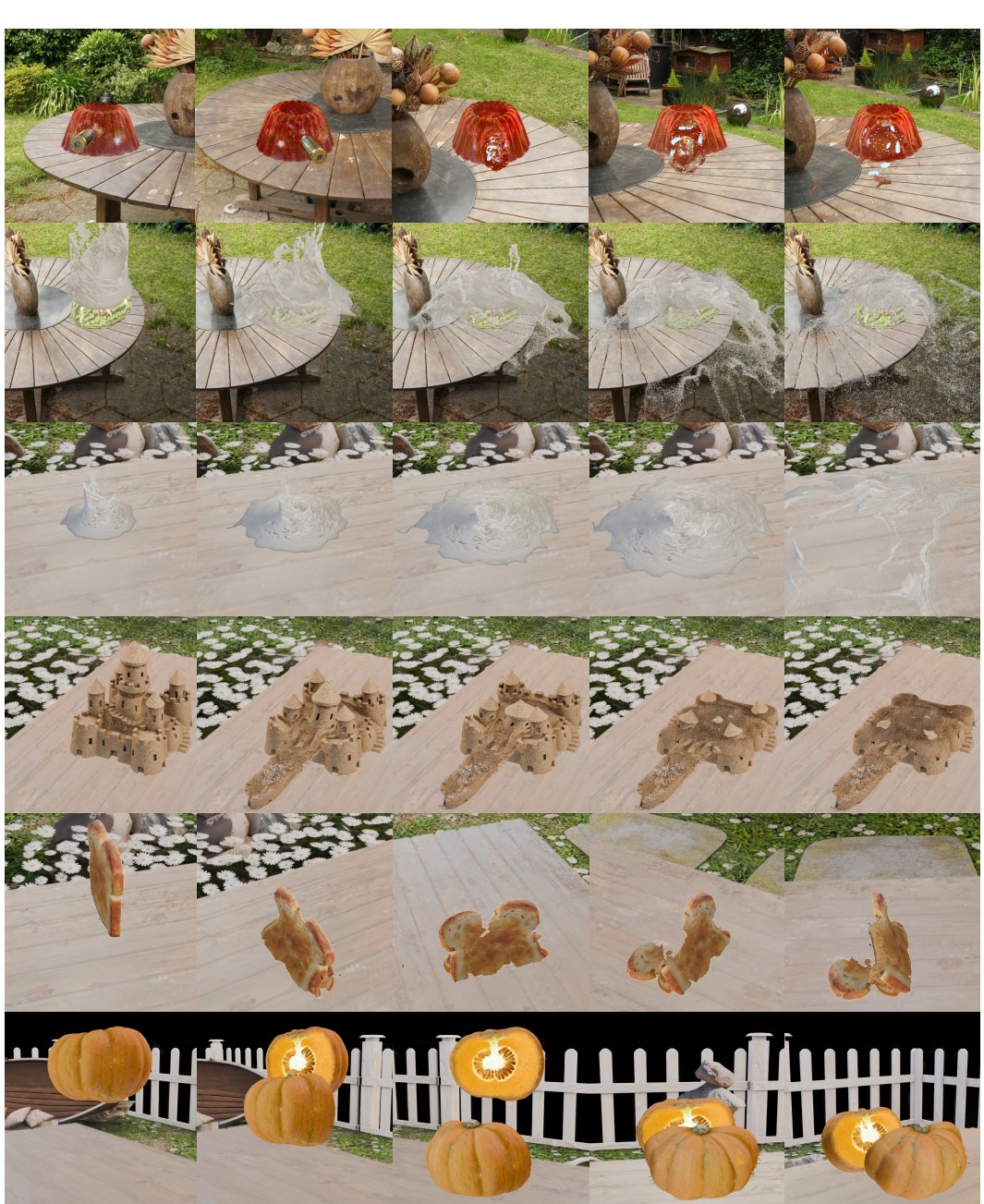

Figure A2: More examples of object simulation.

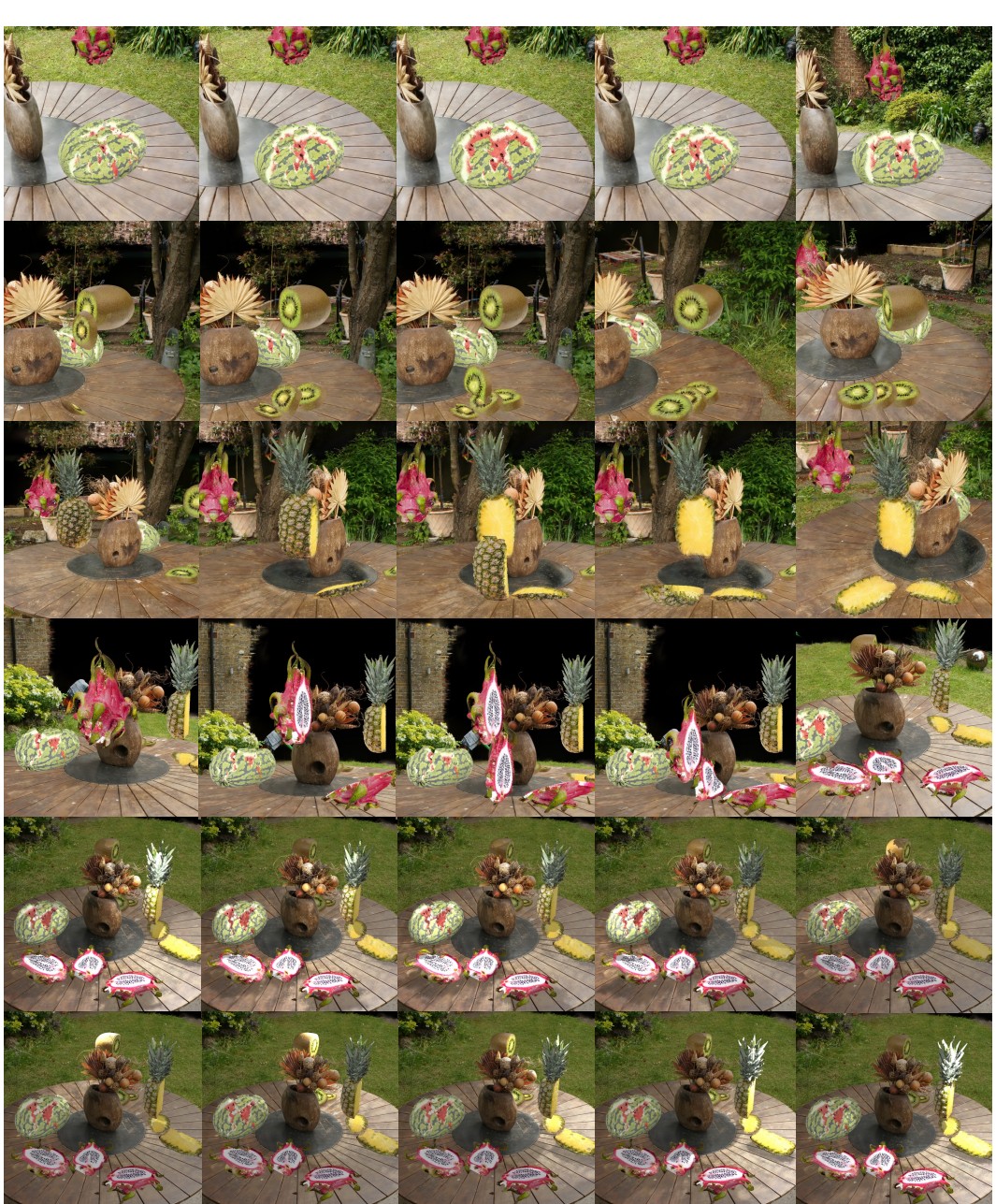

Figure A3: More examples of object simulation and illumination.

