# OpenReview forum: "GaussianFluent: Gaussian Simulation for Dynamic Scenes with Mixed Materials"
_ICLR.cc/2026/Conference — ICLR 2026 Conference Withdrawn Submission_

### Official Review · Reviewer_7ck1 · 2025-10-29

**Soundness:** 3
**Presentation:** 3
**Contribution:** 3
**Rating:** 6
**Confidence:** 4

**Summary:**

This paper introduces GaussianFluent, a framework for realistic simulation and rendering of dynamic object states. The authors synthesize consistent and photorealistic interiors by densifying internal Gaussians guided by generative models. Meanwhile, the pipeline also supports MPM method for simulation and shading. Experiments demonstrate that this method significantly outperforms some baselines.

**Strengths:**

The authors propose a novel pipeline that provides internal structures and textures for 3DGS, which could support more photorealistic simulation. The reviewer finds this part interesting and novel.

The authors introduce the CD-MPM algorithm into 3D-GS and implement the Blinn-Phong lighting model to the framework, supporting deformable simulation and dynamic lighting.

**Weaknesses:**

The proposed internal texture synthesis seems promising. While the reviewer is curious whether the physics information could also be assigned during the texture synthesis.

The inpainting process mainly employs a 2D Diffusion inpainting model, which may cause multi-view inconsistency and affect geometry precision. Why not consider using a 3D diffusion inpainting model? e.g., Amodal3R (Wu et al., ICCV2025)

Section 3.2 seems too heavy. Could the authors reorganize this section for better readability?

**Questions:**

Please refer to weakness.

---

### Official Review · Reviewer_Df6E · 2025-11-02

**Soundness:** 3
**Presentation:** 3
**Contribution:** 3
**Rating:** 4
**Confidence:** 5

**Summary:**

The paper propose a framework to simulate fractures in PhysGaussian framework. To generate fractures, internal textures and a fracture model are needed. To inpaint internal textures, object are cut into slices. Image inpaint is used to generate pixel supervision for internal Gaussian kernels of slices. For fracture simulation, an improved CD-MPM fracture model is proposed. A relighting model is added to enhance realism.

**Strengths:**

- The overall presentation is clear.

- The internal texture inpainting is a good contribution to the PhysGaussian framework. The quality of the generated internal renderings is good. The comparison to previous method in terms of internal rendering is promising.

**Weaknesses:**

- There are many hardcoded parameters. No ablation studies are provided to justify these design choices. What artifacts may appear if these fixed numbers are too large or too small? Some examples:
    - In Iterative Texture Refinement, **5** steps SH optimizations are used for each Gaussian Optimization.
    - **40** slices are used to run inpainting. Is this number suitable for different sizes?
    - **0.1** strength are used in inpainting.

- The shown reconstruction are limited to Objaverse 3D models rendered by blender. Simulatable reconstructions from real-world captures could broad its impact.

- The contribution to the damage simulation model is solid, but seems somewhat outside the topics of this conference.

**Questions:**

- How to make sure the texture across slices are continues? What if some internal parts span multiple slices?

- How to decide inpainting areas? What are the text prompts used for inpainting?

- How to ensure that internal optimization does not affect the surface texture?

- How reflection parameters are determined? How light colors are determined?

- How automated is the Mixed-Material parameter setup? How to get the semantic information of each segmented parts such as seeds, flesh, etc. How to set individual material parameters? Are these steps manual?

- Why does the surface texture of the pineapple at around 1:00 in the video disappear block by block?

- Why are the relighting effects only shown separately in a few demos? Most demos miss shadow casting, which reduces the realism. Are there any constraints to add relighting?

- In region with severe deformations, will spiky artifacts appear? How do you handle overly stretched Gaussian kernels?

---

### Official Review · Reviewer_K1eV · 2025-11-03

**Soundness:** 3
**Presentation:** 2
**Contribution:** 2
**Rating:** 4
**Confidence:** 3

**Summary:**

This paper focuses on learning to simulate dynamic scenes with Gaussian Splatting containing inner structures. The work also aim at learning  physics modeling for the shape. The results demonstrate the effectiveness of the proposed approach.

**Strengths:**

1. The task is crucial for 3D computer vision. The 3DGS only focuses on the learning of surface appearances and ignore the inner structures, which further hinders the capability of physics modeling.
2. The framework is effective by introducing pretrained image models for appearance inpainting.

**Weaknesses:**

1. The method seems to ignore the lighting modeling of the full shape. The demo in teaser seems to only put the Gaussians into a preset scene without adding environment lighting, which looks wired. Since the method also focuses on a "lighting system", solving the environment lighting is also crucial.

2. How dose the rendering-to-inpainting framework keeps the consistency across the slice at different inner structures? For example, if the method is used to learn a physical 3DGS from a cake model, how does the image inpainting model keeps to generate the same cake layer number during inpainting the images across different cake slides?

**Questions:**

Please refer to the weaknesses above.

---

### Official Review · Reviewer_3ggA · 2025-11-05

**Soundness:** 3
**Presentation:** 2
**Contribution:** 3
**Rating:** 4
**Confidence:** 3

**Summary:**

The paper tackles an appealing and timely goal—simulation and rendering of dynamic, mixed‑material scenes within 3D Gaussian Splatting—by combining (i) internal texture synthesis via slice‑wise diffusion‑based inpainting with tri‑axial refinements, (ii) an adapted Continuum Damage MPM (CD‑MPM) with a continuous return‑mapping heuristic and GPU parallelization, and (iii) simple dynamic relighting (Blinn–Phong with PCA normals). While the visuals are engaging and the paper is clearly written, the technical novelty is incremental relative to prior GS+MPM pipelines, several claims (notably “real‑time”) are overstated relative to reported timings, and the evaluation relies heavily on CLIP and small user studies without physics‑grounded validation. The method’s reliance on prompt‑driven interior hallucination plus manual material parameter/segmentation choices weakens scientific rigor and limits reproducibility and generality.

**Strengths:**

- Fig. 2 provides a coherent end‑to‑end pipeline (interior filling → fracture‑aware simulation → relighting), and the text walks through each component with equations and ablations at a readable level

- Identifying/discussing the tip‑projection discontinuity for the NACC return map and proposing a heuristic continuous projection is well motivated, and Fig. A1 effectively illustrates the issue; the write‑up is technically careful.

- The spatially varying  \beta examples (watermelon rind/flesh/seeds) clearly show different fracture behaviors that prior uniform‑material baselines fail to reproduce

**Weaknesses:**

- The abstract and intro emphasize “real‑time” or “real‑time speeds,” yet Table A1 reports 0.39–5.12 s/frame depending on scene, and App. B.3 states a reduction “from 4 minutes per frame to a single second,” i.e., ~1 fps—not real‑time interaction or real‑time rendering in the usual 24–60 fps sense

- Evaluation is largely perceptual/hallucinatory, not physics‑grounded.

- The internal textures are synthesized by 2D inpainting per slice and iterative refinements; while visually convincing (Fig. 6), they are not constrained by material microstructure or tomography and may not be spatially/physically coherent beyond tri‑axial consistency

- Assigning different \beta by color/segments (e.g., seeds/flesh/rind) and holding other parameters fixed is ad‑hoc (Sec. 3.2, “Mixed material simulation”). There is no principled estimation of elastic/plastic parameters from data, nor uncertainty analysis?

- Relighting model is simplistic and weakly validated. (Blinn–Phong with PCA normal)

- Although the paper shows “milk” and “sandcastle” scenes (Fig. A2) and mentions fluids/granular media (Sec. 4.2), the core physics contribution is to brittle fracture via CD‑MPM; no dedicated fluid or granular constitutive models/validations are provided, yet these results are used as evidence of breadth.

- Key hyperparameters lack sensitivity analyses. The effect of the scale regularizer on reconstruction fidelity vs. normal quality is not quantified

**Questions:**

See above in the weakness section.

---

### Note · Authors · 2025-11-14

I have read and agree with the venue's withdrawal policy on behalf of myself and my co-authors.